# An Open Multifunctional FPGA-Based Pulser/Receiver System for Intravascular Ultrasound (IVUS) Imaging and Therapy

**DOI:** 10.3390/s25154599

**Published:** 2025-07-25

**Authors:** Amauri A. Assef, Paula L. S. de Moura, Joaquim M. Maia, Phuong Vu, Adeoye O. Olomodosi, Stephan Strassle Rojas, Brooks D. Lindsey

**Affiliations:** 1Graduate Program in Electrical and Computer Engineering (CPGEI), Federal University of Technology—Paraná (UTFPR), Curitiba 80230-901, PR, Brazil; paulamoura@alunos.utfpr.edu.br (P.L.S.d.M.); joaquim@utfpr.edu.br (J.M.M.); 2Department of Biomedical Engineering, Georgia Institute of Technology, Atlanta, GA 30332, USA; pvu40@gatech.edu (P.V.); aolomodosi3@gatech.edu (A.O.O.); ststrassle@gatech.edu (S.S.R.); brooks.lindsey@bme.gatech.edu (B.D.L.)

**Keywords:** intravascular ultrasound, ultrasound electronics, open system, FPGA

## Abstract

Coronary artery disease (CAD) is the third leading cause of disability and death globally. Intravascular ultrasound (IVUS) is the most commonly used imaging modality for the characterization of vulnerable plaques. The development of novel intravascular imaging and therapy devices requires dedicated open systems (e.g., for pulse sequences for imaging or thrombolysis), which are not currently available. This paper presents the development of a novel multifunctional FPGA-based pulser/receiver system for intravascular ultrasound imaging and therapy research. The open platform consists of a host PC with a Matlab-based software interface, an FPGA board, and a proprietary analog front-end board with state-of-the-art electronics for highly flexible transmission and reception schemes. The main features of the system include the capability to convert arbitrary waveforms into tristate bipolar pulses by using the PWM technique and by the direct acquisition of raw radiofrequency (RF) echo data. The results of a multicycle excitation pulse applied to a custom 550 kHz therapy transducer for acoustic characterization and a pulse-echo experiment conducted with a high-voltage, short-pulse excitation for a 19.48 MHz transducer are reported. Testing results show that the proposed system can be easily controlled to match the frequency and bandwidth required for different IVUS transducers across a broad class of applications.

## 1. Introduction

Coronary artery disease (CAD), which can result in cardiac ischemia, is the third leading cause of disability and death worldwide, associated with more than 17.8 million deaths every year [1]. CAD involves the development of complex, chronic atherosclerosis, an inflammatory disease characterized primarily by a progressive accumulation of plaque within the artery, which affects normal blood flow to the myocardium [2]. Recent studies have shown that several biomechanical indicators associated with vulnerable plaque rupture can be useful for the risk stratification of stable CAD [1,3]. These indicators include the influence of stenosis geometry [3,4], plaque composition (e.g., thin cap fibroatheroma) [5], morphology [6], and hemodynamics [7], including wall shear stress [8], among others [9,10]. Despite rapid scientific and technological advances, the relationship mechanisms between the primary factors driving plaque vulnerability (i.e., likelihood of plaque rupture) are still incompletely understood and continue to pose a challenge for gaining a comprehensive view of atherosclerotic plaque [1,11]. Therefore, new approaches based on simultaneously measuring the established biomechanical markers of vulnerable plaques in vivo may improve the understanding of CAD progression and aid in diagnosis, disease management, and treatment strategies [12].

Several imaging techniques have been used to assess the plaque structure and stenosis severity with high resolution and contrast [13]. Coronary angiography is a minimally invasive anatomical imaging modality considered the gold standard for diagnosing CAD [14]. Catheter-based angiography is typically performed in specialized cardiac catheterization laboratories using X-ray imaging and a catheter-directed intra-arterial contrast agent to enhance the visibility of the blood vessels [15]. However, several studies have shown the limitations of catheter-based angiography, including its limited ability to assess the full extent of plaque accumulation due to the 2D projection imaging of 3D anatomy, and the corresponding overestimation or underestimation of the severity of atherosclerotic disease burden [16,17]. Therefore, for patients already in the cardiac catheterization lab, intravascular imaging, such as intravascular ultrasound (IVUS), is recommended to characterize the stenosis [18,19]. IVUS imaging is a well-known, minimally invasive imaging modality that uses radiofrequency (RF) backscattered ultrasound signals to generate high-resolution images of vascular structures [20], particularly during percutaneous coronary intervention (PCI), when a stent is placed in the stenotic segment of the artery [21]. Conventional side-viewing IVUS provides qualitative 2D cross-sectional imaging of coronary anatomy, information which may be valuable because the extent and structure of atherosclerotic plaques can be directly imaged during the procedure [22]. However, its application is limited to non-occlusive CAD due to the local geometry constraint, making its use not feasible in complex lesion interventions, such as coronary chronic total occlusion [23]. In these cases, forward-viewing IVUS has the additional volumetric blood flow imaging capability for guiding interventions and extracting color flow measurements and analysis [24].

A critical component of an IVUS system is the ultrasound transducer. In recent years, significant advances in novel ultrasound transducers for both side-viewing and forward-viewing IVUS applications have been reported by several groups [25,26], including ours [27,28,29], for characterizing plaque composition, morphology, local lesion structure, and hemodynamics in real-time. When developing a new IVUS transducer or system, in vitro, ex vivo, and ultimately in vivo experimental testing of the new device or system may commonly utilize a research ultrasound platform, such as the Verasonics Vantage System (Verasonics, Redmond, WA, USA) [30,31]; classical instrumentation, such as arbitrary waveform generators, RF power amplifiers, digital oscilloscopes, pulser/receiver systems, and data acquisition boards; or custom system electronics developed at the researchers’ institution [32,33,34,35,36]. However, such approaches have challenges, particularly for IVUS, including their cost, large size and bulky form factor, and the difficulty of incorporating them into a compact, minimally invasive system, limiting their flexibility and portability [33]. As a result, several groups have proposed novel ultrasound systems to support the development of new and effective IVUS methods.

Qiu et al. [37] proposed an open ultrasound system specifically designed for IVUS capable of generating bipolar pulses with a maximum amplitude of 160 Vpp. The same group also employed the system to evaluate a novel dual-mode imaging catheter with both a forward-viewing transducer (22.5 MHz) and a side-viewing transducer (39.1 MHz) for simultaneous high-resolution morphological and functional imaging in CAD [38]. Qiu et al. also developed an open ultrasound system for modulated-excitation-based IVUS imaging using an arbitrary waveform generator [39]. The system could generate linear high-voltage coded excitation waveforms from 20 MHz to 80 MHz, with a peak amplitude of 60 Vpp, and could digitize the analog signals with a maximum sampling rate of 250 MSPS. A 1 GSPS digital-to-analog converter (DAC) and a two-stage power amplifier were incorporated for pulse generation. Quantitative measurements using chirp-coded excitation demonstrated an improvement of 12% in the SNR, while the penetration depth increased by 47.1% and 86.7% for 30 MHz and 60 MHz IVUS transducers, respectively. Alternatively, Liu et al. [40] reported a dual-channel real-time imaging system to evaluate a dual-element imaging catheter. In this approach, two 40 MHz transducer elements with similar performances were arranged in a back-to-back configuration to improve nonuniform rotational distortion (NURD) in mechanically steered IVUS. The dual-channel pulse generator circuit could generate bipolar pulse excitation with a peak amplitude of 94 Vpp using a pair of high-voltage and high-current metal oxide semiconductor field effect transistors (MOSFETs). More recently, Liu et al. [41] introduced an ultra-high-frequency (>100 MHz) coded excitation imaging platform to improve the imaging resolution and signal-to-noise ratio (SNR) for IVUS imaging at these high frequencies. The platform consisted of a high-performance transmitter, which could generate single and coded excitation pulses with an amplitude exceeding 50 Vpp, and a high sampling rate signal receiver (500 MSPS). These systems are helpful for investigating new IVUS imaging methods; however, they require complex and expensive high-density multilayer (up to 12 layers) printed circuit board (PCB) technology for implementation. Additionally, these studies demonstrate that the minimum adjustable center frequency is 10 MHz and the pulse excitation scheme supports only two-level bipolar non-return-to-zero (NRZ) sequences (duty cycle of 100%).

Separately from these custom systems, available commercial ultrasound machines do not always provide extensive user control of the transmission (TX) and reception (RX) parameters, as well as direct access to raw RF signals during pulse-echo measurements [30]. To enable the evaluation and characterization of custom IVUS transducers, including the effect of varying pulsing parameters, a flexible and open ultrasound research platform with a dedicated electronic system is required. It would also be desirable if such an open development system enabled researchers to easily reprogram reconfigurable hardware technologies, such as Field-Programmable Gate Arrays (FPGAs), through a user-friendly interface for accurate waveform control and data acquisition.

This paper presents the development of a novel FPGA-based multifunctional and fully reconfigurable pulser/receiver (P/R) ultrasound system for IVUS imaging and therapy research. The system features a method for encoding complex waveforms using the pulse width modulation (PWM) technique at a sampling frequency of 200 MHz, suitable for a three-level return-to-zero (RTZ) ultrasound pulser. The system is based on an FPGA board and a proprietary PCB for compact, flexible, and cost-effective implementation. To our knowledge, this is the first portable, fully reconfigurable system for forward-viewing IVUS imaging and therapy with the ability to generate both three-level conventional and complex coded excitation using a center-aligned PWM scheme with a programmable duty cycle. The proposed dedicated system enables the implementation of new ultrasound transmission or reception approaches, including the signal and image processing investigation methods required to develop novel forward-viewing IVUS transducers for various imaging modes (color flow, shear wave elasticity imaging) and therapeutic applications (e.g., intravascular thrombolysis). In particular, the ability to produce long pulses between 100 kHz and 25 MHz that are optimized to the particular IVUS transducer and imaging or therapy mode is unique to this system.

This paper is organized as follows: First, the overall system architecture is described in detail in Section 2. Experiments were conducted to characterize the system with long and short excitation pulses, and the results are presented in Section 3, which is followed by the discussion provided in Section 4. Finally, the conclusions are presented in Section 5.

## 2. System Description

The block diagram of the proposed system is illustrated in Figure 1. The fully reconfigurable and open platform for IVUS research consists of a host PC, a commercial FPGA development board, and a compact proprietary TX/RX analog front-end (AFE) board. The digital hardware architecture is connected via a USB 2.0 interface to the computer, in which a Matlab-based software application provides a flexible and easy-to-use interface for three-level arbitrary waveform generation, digital hardware control, and direct raw RF ultrasound data acquisition.

An Altera DE2-115 board (Terasic Inc., Hsinchu, Taiwan) was used for timing, waveform, and spectrum characteristic control, command, and data acquisition. This board uses the Cyclone IV E EP4CE115F17C6N FPGA (Intel Corporation, Santa Clara, CA, USA), which contains a programmable 114,480 logic elements (LEs), 3.9 Mbits of RAM, 266 embedded multipliers, and four phase-locked loops (PLLs). The board also includes a set of peripherals and interfaces, such as the 172-pin High-Speed Mezzanine Card (HSMC), which meets the high bandwidth requirement to interface with the custom proprietary board. The Intel Quartus Prime 15.1 Lite Edition was selected for high-level block design and the digital synthesis of VHDL code, and the Nios II 15.1 IDE for synthesizing C functions into register-transfer level (RTL) hardware description language (HDL). The design code was tested using the ModelSim-Altera 10.4b simulator.

The four-layer custom board had an overall size of 81 × 87 mm and included both the TX and RX front-end electronics requested for a large class of ultrasound research. The main functions of the AFE board were to receive commands and digital pulse code data for the waveform generation required for the piezoelectric transducer (XDCR) excitation and to amplify and sample the received echo signals during pulse-echo experiments. This task was performed by a wideband voltage-controlled amplifier (VCA) that varied its gain depending on an analog control voltage provided by a serial DAC and two OP-AMP circuits. Therefore, strong echo signals reflected from vascular tissues and weak signals backscattered from blood could be detected with the appropriate time gain compensation control. A differential amplifier (DIF AMP) converted a single-ended RF signal into a differential signal to meet the ADC input requirements. The incoming data digitized by the high-speed ADC was then transferred to the computer for post-processing, visualization, and analysis. Nine SubMiniature version A (SMA) connectors were available to insert single-element transducers: one directly connected to the pulser output and the other eight connected to a high-voltage (HV) multiplexer (MUX), allowing the single ultrasound pulser to drive up to eight transducer elements.

### 2.1. Transmitter

The high-speed and high-voltage transmitter used in the proposed prototype is the one-channel ultrasound pulser HV7361 (Microchip Technology, Chandler, AZ, USA). The HV7361 is a three-level 200 Vpp (±100 V) pulser with peak output currents up to 2.5 A, which makes it suitable for high-voltage IVUS applications. The pulser is composed of four controller logic interface circuits; four level translators; four AC-coupled MOSFET gate drivers, powered by an external power supply of 10 V; and two pairs of high current complementary P-channel and N-channel MOSFETs connected as two push–pull output stages to excite the transducer. One push–pull driver is fed by two power supplies referenced to VPP (positive) and VNN (negative), ranging from 0 V to +100 V and from 0 V to −100 V, respectively. The other push–pull circuit is connected to 0 V for damping control, allowing for fast RTZ output of the three-level waveforms applied to the transducer. As an additional feature, the pulser includes an integrated two-terminal, low-noise T/R switch that isolates the receiver AFE from the high-voltage pulses produced by the transmitter during the TX operation. The application circuit of the HV7361 and its logic control table, including the control logic input pulse pins and the output MOSFETs, are shown in Figure 2 and Table 1, respectively.

From Table 1, the FPGA is programmed to control the amplitude and frequency of the pulsed excitation waveforms using a symmetrical PWM with one pulse per half-cycle. This technique is particularly advantageous for implementing three-level conventional and coded excitation signals due to the inherent low-pass filtering of the transducer output signal [42], which effectively integrates the bipolar pulses. The digital PWM technique is characterized by alternating ON and OFF pulses in a specific pattern to represent an analog rectangular waveform by varying the duty cycle (*D*). Considering the active pulse on-time $t_{ON}$ and off-time $t_{OFF}$, the duty cycle is expressed as follows:(1)$$D=\frac{t_{ON}}{t_{ON}+t_{OFF}}=\frac{t_{ON}}{T},$$
where *T* is the period of the excitation waveform determined by the spacing between two periodic pulses of the same polarity.

This work exploits the center-aligned PWM method over each half-cycle between zero-crossings to convert arbitrary analog waveforms into tristate pulses. A positive half-cycle for a 5 MHz (*T* = 200 ns) transmit frequency and a duty cycle value of 0.5 (50%) is shown in Figure 3 as an example to illustrate the proposed approach. In this method, the center-aligned carrier signal (triangular wave) aligns the rectangular PWM output signal to the peak of the original reference signal, which serves as the reference point. Then, the duty cycle modulated signal is compared to the triangular carrier wave: if the carrier wave is higher than *D*, the PWM pulse is high (ON state—logic level 1), while if the carrier wave is lower than *D*, the PWM pulse is low (OFF state—logic level 0). This result is quantized at a sampling frequency of 200 MHz (i.e., interval time of 5 ns) to set the pulse width with 20-time samples. The proposed algorithm is implemented to maintain symmetry between successive positive and negative rectangular pulses, which can be evaluated independently.

Assuming that the width of the rectangular pulse is used to encode the amplitude of the excitation waveform during a half-cycle, Figure 4 shows examples of the simulation to demonstrate qualitatively this approximation for a 2-cycle 5 MHz pulse with the duty cycle varying from 0.25 to 1.0 in 0.25 increments. The tristate drive signals (transmitter waveforms) to be applied by the pulse generator with the duty cycle variation, where two consecutive rectangular pulses with the same polarity define the frequency of the output signal, and their decomposition into the positive (INA), negative (INB), and zero (INC) control levels, are shown in Figure 4a and Figure 4b, respectively. The behavior of the amplitude of the fundamental frequency component of the wave, which is a consequence of the Fourier decomposition of the rectangular pulse, is illustrated in Figure 4c. In this case, the amplitude A of the fundamental frequency components, obtained by utilizing Equation (2) (adapted from [42]), with a maximum value of 1/π for *D* equal to 1, is normalized for better visualization.(2)$$A=\frac{1}{T}\int_{t_{0}}^{t_{0}+T}f\left(t\right)dt=2.\frac{1}{T}\int_{0}^{\frac{D.T}{4}}cos(2\pi ft)dt=\frac{sin\left(2\pi f\frac{D.T}{4}\right)}{\pi }$$

After the pulse definition and simulation, the input control signals for each half-cycle are compressed into a 16,384 × 16 data buffer to save the memory resources of the FPGA. The 16-bit array implemented to save the bipolar three-level code sequence is defined in Table 2, where the two most significant bits are used to specify the pulse level, while the subsequent bits are reserved for storing the number of 5 ns clock periods for each level.

Figure 5 shows a complete example of a two-cycle, 10 MHz sinusoidal pulse conversion into the drive signals required for the HV7361 pulser. First, the user specifies the waveform parameters, such as type, frequency, pulse duration, duty cycle, and window function (Figure 5a) in the software interface. Then, the custom algorithm converts this specification into a pulse code data sequence that is compressed and stored in a 16k word buffer (Figure 5b). Finally, the computed array data is transferred from the host computer to the FPGA, which decompresses the tristate sequence to be applied to the pulser with a precise pulse width using the PWM method with a 200 MHz resolution (Figure 5c).

### 2.2. Receiver

Figure 6 shows the simplified block diagram of the RX AFE. The first module of the receiver segment was the T/R switch, which was integrated into the HV7361 pulser. The T/R switch isolated the HV excitation pulses that drove the ultrasound transducer from the low-voltage (LV) and sensitive amplification electronics. Following the T/R switch, a low-VCA810 (Texas Instruments, Dallas, TX, USA) noise automatic variable gain amplifier, which adjusted its gain based on an external reference voltage, was chosen as the VCA. It had a –3 dB gain control bandwidth of 25 MHz and achieved a linear dynamic range gain control from −40 dB to +40 dB by controlling the gain control input pin (Vc) from 0 V to −2 V, respectively, with an input voltage noise density of 2.4 nV/√Hz at 25 °C. To increase the flexibility of the variable gain amplifier, a low-power and single-channel digital-to-analog converter, DAC7311 (Texas Instruments, Dallas, TX, USA), which communicated with the FPGA via an SPI interface at a clock rate of 50 MHz, was adopted for setting the gain control voltage reference. This 12-bit serial DAC allowed linear-in-dB continuous gain control for time-gain compensation. To avoid damage to the VCA, the signal coming from the DAC, which operated at 3.3V, was passed through an amplification stage using two of the Op Amp OPA211 (Texas Instruments, Dallas, TX, USA) set to a fixed gain of −0.68 (i.e., maximum voltage of −2.2 V).

A second gain stage of +10 dB and bandwidth of 1900 MHz using the Op Amp THS4509 (Texas Instruments, Dallas, TX, USA) was included in the hardware design. The wideband and fully differential operational amplifier with a low noise of 1.9 nV/√Hz and very low harmonic was used to convert the single-ended output of the VCA810 to a pair of differential signals required for the high-speed ADC, represented by the 12-bit 105 MSPS sampling frequency ADS6124 (Texas Instruments, Dallas, TX, USA). Before digitization, the output of the differential amplifier was filtered through a low-pass anti-aliasing filter (AAF) with a –3 dB cutoff frequency of 40 MHz to prevent the aliasing of out-of-band noise and interfering components. The implementation of the AAF circuit was based on the Texas Instruments ADS61xx evaluation module [43]. The digital data outputs of the ADC were connected to the FPGA via a parallel interface. The standard drive circuit connection between the differential amplifier and the ADC input pins (INP and INM) was AC-coupled for optimum performance. In this case, the INP and INM input pins were externally biased around the common-mode output voltage of 1.5 V available at the output VCM pin. Therefore, the differential lines of the input signal swung symmetrically between VCM + 0.5 V and VCM—0.5 V (differential input signal swing of 2 Vpp).

Another feature of our ultrasound system was the possibility of individually evaluating up to eight transducer elements using HV analog switches. For this application, we used the HV20200 (Microchip Technology, Chandler, AZ, USA) integrated circuit, an eight-channel HV MUX with an output on-resistance of 22 Ω controlled by an SPI interface. Each channel output was connected to a 50 Ω SMA connector to achieve enhanced impedance matching and signal integrity between the MUX and the transducer. On-board RC parallel equivalent circuits could be connected/disconnected to each channel as test loads using jumpers.

### 2.3. Customized Software Graphical User Interface (GUI)

The GUI used to reconfigure and control the hardware system was developed using the MATLAB^®^ software version 2022a through the App Designer tool, on a personal computer powered by an Intel^®^ Core^TM^ i5 (2.0 GHz) CPU and 16 GB of RAM. As illustrated in Figure 7, it allows for easy user interaction with the digital architecture by selecting one of the four excitation schemes and the excitation parameters, providing access to raw echo data for echo signal processing. The software converts the chosen parameter settings into the corresponding commands and synthesized pulse code, and then transfers them to the FPGA device, which acts as a control processor unit, as described in Section 2.1. The first excitation scheme is conventional sinusoidal excitation, the traditional and most widely used method for ultrasound pulse excitation [44]. The other schemes include chirp-coded excitation [45] (maximum duration of 1 ms), Golay complementary sequence [46] (length of 4-, 8-, or 16-bits), and Barker sequence [47] (length of 2-, 3-, 4-, 5-, 7-, and 11-bits), which can be used to improve the SNR and the penetration capability with the application of more sophisticated signal processing techniques [48,49]. Once the user has selected the excitation method and the appropriate parameters, such as pulse frequency, duty cycle, number of cycles, and polarity, a preview of the pulse code generation is shown in the “Graphics” window after pressing the “Generate” button. One example of a three-cycle 5 MHz sinusoidal pulse with a Hanning window, damping enabled, and a duty cycle of 70% is shown in Figure 7. The waveforms on the “Graphics” window show the simulated waveform, the converted tristate signal, and the three output data signals that will be compressed and sent to the FPGA for pulser control by pressing the “Send to FPGA” button. The “Start US Tx-Rx” button starts the pulse-echo RF acquisition with the following parameters: single pulse or pulse repetition frequency (PRF) from 1 Hz up to 5 kHz.

### 2.4. Hardware Characterization

Before the excitation experiments, the RX front-end amplification circuit was characterized by gain linearity and frequency response using the TINA-TI simulation tool (DesignSotf, Budapest, Hungary) and an experimental setup for comparison. For experimental measurements, a single-channel arbitrary function generator (AFG3021, Tektronix, Beaverton, OR, USA) was employed to generate a small sine wave, as a signal source, which was applied to the transducer input connector to simulate the echo. Then, the system performance of generating analog waveforms was evaluated using two custom single-element piezoelectric transducers with center frequencies of 550 kHz and 19.48 MHz, which were recently developed by our group [28,29]. A high-voltage DC/DC power supply, AN-H59DB1 (Microchip Technology, Chandler, AZ, USA), provided the VPP and VNN voltages needed for the pulser (up to ±90 V). The voltage waveforms were recorded by a 2.5 GSPS digital oscilloscope (MDO3014, Tektronix, Beaverton, OR, USA).

First, an example of multicycle therapy excitation is presented. The acoustic output parameters of the 550 kHz transducer with 1000 cycles were measured by a calibrated hydrophone NH-0200 (Precision Acoustics, Dorchester, UK) positioned at 2 mm from the transducer surface in a degassed water tank (40 × 30 × 25 cm). To map the spatial distribution of the acoustic pressure profile, the transducer was fixed on a holder, and the hydrophone was attached to a computer-controlled motion stage (Newport XPS, Irvine, CA, USA), allowing 2D mechanical scanning along the acoustic propagation axis. The acoustic parameters to compute the deposited energy in-water measurement were estimated as in [50], including the derated spatial-peak temporal-average intensity (*I_SPTA.3_*), the derated spatial-peak pulse-average intensity (*I_SPPA.3_*), and the mechanical index (MI). The thermal performance of the system with the long-duration pulse was also monitored using an FLIR T540 infrared thermal imaging camera (Teledyne FLIR LLC, Wilsonville, OR, USA), which has a resolution of 464 × 348 pixels (161,472 measurement points). Second, a pulse-echo experiment was conducted to evaluate the performance of the P/R system in generating a high-voltage short pulse for high-resolution ultrasound imaging. In such a test, a 75 µm diameter tungsten wire was placed 5 mm from the 19.48 MHz transducer. Finally, examples of multicycle arbitrary waveform generation were presented.

## 3. Results

Figure 8 shows photographs of the implemented multifunctional hardware architecture. Figure 8a shows the Altera DE2-115 FPGA board connected to the AFE board. Figure 8b shows a photo of the compact AFE board with detailed labeling of the main blocks required to investigate highly flexible TX/RX IVUS strategies.

### 3.1. Characterization of the Receiver Circuit

Figure 9 shows the schematic circuit to simulate the performance of the receive segment. The output of the VCA810 is the signal VCAout, which is connected to the differential amplifier THS4509. The control voltage produced by the DAC is represented by a fixed voltage source (Vin_0dB), in this case, 1.47 V for 0 dB gain. The conditioned output signal of the VCA810 (VCA_out) is applied to the THS4905 circuit. This process is essential to ensure the correct implementation of individual circuits and integration of the AFE.

Figure 10 presents the comparison between the simulated and the compensated experimental gain curve response to control the VCA device through the analog control voltage (Vc). The values obtained experimentally are updated in the software interface to control the gain of the RF amplifier from −40 dB to +40 dB with higher precision and linearity.

Figure 11 allows for the analysis of the gain linearity by plotting the small-signal frequency response curves of the receiver circuit for the gain values from −20 dB to +20 dB with increments of 10 dB. The input signal used to characterize the circuit in both simulated (Figure 11a) and measurement results (Figure 11b) is a sine wave with a frequency range from 100 Hz to 25 MHz and amplitude varying from 50 mV to 500 mV. The gain linearity is satisfactory in all cases with less than 3 dB fluctuation, which is especially relevant in the frequency range of this study.

### 3.2. Performance of the Therapeutic Ultrasound System

The system was characterized to demonstrate its ability to generate long pulses for therapeutic ultrasound applications. A 1000-cycle 550 kHz pulse with a Tukey window (25% taper ratio) was applied to a custom 3 × 3 mm transducer for mapping the acoustic field based on hydrophone measurements. Detailed descriptions of the therapy transducer were published previously by our group [28]. Figure 12 shows the simulated signals generated by the software interface. The first signal in Figure 12a (upper panel) is the 1000-cycle waveform with windowing. This waveform was converted to the tristate bipolar signal computed with a duty cycle of 70% (middle panel) to obtain the input control signals to the pulser (lower panel). Due to the long excitation period (≈1.82 ms), the time duration from 800 µs to 820 µs in Figure 12a was enlarged in Figure 12b for better illustration.

The experimental digital input control signals INA (channel 1), INB (channel 2), and INC (channel 3), and the analog output three-level pulse (channel 4) with a peak amplitude of 72 Vpp can be seen in Figure 13a. A period of 20 µs along the 2 ms recorded signal in the oscilloscope marked with a black rectangle is magnified for a more detailed view. The calculated spectrum of the output pulse using the Fast Fourier Transform (FFT) algorithm is shown in Figure 13b, indicating the fundamental frequency component of 549.32 kHz, which has a −6 dB absolute bandwidth of 476.84 Hz, i.e., a fractional bandwidth of 0.09%. In this experiment, the power supply was set to 70 Vpp.

The experimental beam plots presented in Figure 14 were obtained using the hydrophone with the same parameters described above, with a PRF of 2 Hz. A square plane with an area of 4 × 4 mm^2^ was scanned with a step size of 0.1 mm in each direction. The system recorded the acoustic signals and converted them to pressure values using the hydrophone transfer function provided by the manufacturer to construct the peak negative pressure map. Figure 14a shows the 2D spatial distribution of the acoustic gain produced by the transducer, which is converted to a peak negative acoustic pressure map in Figure 14b. The 2D contour beam plots of ultrasound intensities *I_SPTA.3_* and *I_SPPA.3_* for the 550 kHz transducer with a peak centered in (0, 0, 2 mm) are shown in Figure 14c and Figure 14d, respectively. During the acoustic characterization, the hydrophone-measured peak negative acoustic pressure was 1.8 MPa, resulting in a sensitivity of 0.025 MPa/Vpp. The derated *I_SPTA.3_* was measured as 39.05 mW/cm^2^, while the derated *I_SPPA.3_* was 10.74 W/cm^2^. The corresponding MI was 2.41. The beam produced by the transducer exhibited a −3 dB beamwidth of approximately 1 mm. The VCA810 was adjusted for a gain of +20 dB to amplify the reflected signals, with the ADC operating at a sampling frequency of 100 MHz.

The thermal performance of the proposed system during prolonged use was evaluated with the same excitation sequence and PRF of 200 Hz. Figure 15 illustrates the thermal image using the FLIR infrared camera positioned 10 cm from the AFE PCB. The experiment was conducted at an ambient temperature of 22 °C with no forced cooling. The maximum steady-state temperatures recorded were 66.6 °C at the VCA810 and 51.6 °C at the ADS6124, respectively. In contrast, the HV7361 pulser achieved a maximum temperature of 32.7 °C. These temperatures were below the 70° C limit for commercial electronics.

### 3.3. Performance of the Ultrasound Imaging System

The system’s performance in generating a high-voltage short pulse for high-frequency ultrasonic transducers was evaluated using conventional excitation with 20 MHz central frequency, a duty cycle of 100%, a one-cycle duration, and a rectangular window function. The power supply was adjusted to 90 V for the VPP and VNN voltages (180 Vpp). Table 3 summarizes the main properties of the piezoelectric transducer used in this evaluation, which is similar to the transducer presented in [51]. Photos of the side and top views of the custom transducer are shown in Figure 16a and Figure 16b, respectively.

In Figure 17a,b, a monocycle 181 Vpp bipolar pulse and its spectrum are shown, respectively. The −6 dB cutoff frequencies are 5.88 MHz and 34.08 MHz, resulting in an experimental absolute bandwidth of 28.2 MHz (fractional bandwidth of 141%). The pulse presented in this subsection was applied to a 0.8 mm diameter, 20 MHz transducer for pulse-echo testing. Figure 18 plots the reflected echo signal sampled at 100 MSPS and its spectrum from the tungsten wire at a focal distance of 5 mm from the transducer face. The experimental central frequency is 19.48 MHz with a −6 dB bandwidth of 7.52 MHz (i.e., fractional bandwidth of 38.59%) and an SNR of 26.01 dB.

### 3.4. Examples of Arbitrary Pulse Generation

Three different examples are provided in Figure 19 to demonstrate the capability of the prototype system of generating arbitrary waveforms, which are typically employed to improve SNR and penetration characteristics: a linear-frequency-modulated (LFM) chirp excitation, a Golay complementary sequence, and a Barker code.

Figure 19a shows the input signals INA (channel 1), INB (channel 2), and INC (channel 3), and a 69 Vpp chirp signal (channel 4) parameterized with the modulation frequency from 4 MHz to 11 MHz with a 7 µs duration and a duty cycle of 100%. The chirp-coded pulse was generated with a 40% tapered Tukey window, resulting in a −6 dB bandwidth of 3.5 MHz, obtained from its spectrum in Figure 19b. The same sequence of images is presented in the following figures. The second example that could be applied to achieve a higher SNR in modulated excitation was a 16-bit Golay sequence with a one-cycle bit length and a duty cycle of 50%. The programmable data sequence was +1, +1, +1, −1, +1, +1, −1, +1, +1, +1, +1, −1, −1, −1, +1, and −1, in which the value +1 referred to a cycle with positive phase and −1 designated a cycle with negative phase. The peak amplitude of the output pulse waveform in Figure 19c is 78 Vpp with a center frequency of 10 MHz. Its calculated pulse spectrum is shown in Figure 19d. The third strategy employed to enhance the image quality by increasing the time-bandwidth product of the transmitted pulse was Barker-coded excitation. Figure 19e illustrates a 78 Vpp 10 MHz Barker sequence with a length of 13 and a duty cycle of 50%. The sequence of concatenated positive and negative polarities to produce the tristate-driven Barker-coded pulse was given by +1, +1, +1, +1, +1, −1, −1, +1, +1, −1, +1, −1, and +1. Its spectrum is shown in Figure 19f.

### 3.5. Signal Integrity and Noise Floor Experiment

The frequency spectrum of the reconstructed signal was used to analyze the ADC performance for input signals with low and high frequencies (*f_in_*) at the sampling frequency (*f_s_*) of 100 MSPS. The arbitrary function generator was used to generate the input sine wave signals for the experiments with an amplitude of 0.4 Vpp at frequencies of both 550 kHz and 20 MHz. The power spectra for an input frequency of 550 kHz are presented in Figure 20a−d for the following specifications: (a) spurious free dynamic range (SFDR), (b) signal-to-noise-and-distortion ratio (SINAD), (c) SNR, and (d) total harmonic distortion (THD). The frequency scale was limited to 25 MHz (i.e., half the Nyquist frequency) for the better visualization of the frequency components. However, the data existed for the entire Nyquist bandwidth (from DC to *f_s_*/2).

The power spectrum of the signal in Figure 20a displays an SFDR of 43.44 dBc (relative to the carrier) with the annotation “F” indicating the fundamental (carrier) frequency of 550 kHz and “S” indicating the spur level at the first harmonic (1.1 MHz). Figure 20b displays the SINAD of 40.08 dB. This parameter was a strong indicator of the overall performance of an ADC, because it was calculated as the ratio of the total signal to the sum of all distortion and noise components. Figure 20c shows the SNR of 43.24 dB and the annotation of the first six harmonics. The SNR was calculated similarly to SINAD, except that the signal harmonics were excluded from the calculation, leaving only the noise terms. Figure 20d reports the THD of −42.92 dB, determined from the fundamental frequency and the first five harmonics. The calculated noise floor was −96.39 dB. More details about the specifications to quantify the distortion and noise of an ADC can be found in [52].

The power spectra for a signal with an input frequency of 20 MHz are shown in Figure 21a−d for the same specifications described above over the entire Nyquist bandwidth. The SFDR, SINAD, SNR, and THD were 39.40 dBc, 38.29 dB, 44.76 dB, and −39.40 dB, respectively. The calculated noise floor was −91.92 dB. There was a slight degradation of approximately 4 dB in SFDR. As expected, a slight degradation occurred at higher frequencies for both SINAD and SNR due to several factors, including sampling jitter, clock jitter, and the increased slope of the input signal at higher frequencies, which can lead to more sampling errors [53].

### 3.6. FPGA Resource Utilization

Table 4 presents the FPGA resource usage summary obtained by synthesizing the proposed hardware architecture using Quartus Prime 15.1. Even using a 32-bit embedded processor Nios II to control the communication system between the computer and the FPGA, the design used less than 15% of the hardware resources, except for the PLL. This result indicated that the embedded system could include new algorithms and functionalities.

## 4. Discussion

Despite the advances over the past decades in IVUS-based imaging modalities, methodological challenges and the lack of quantification metrics for risk stratification have motivated the research into novel approaches to improve the accuracy of diagnostic and therapy applications [12,22,54]. However, the available commercial systems do not always meet the needs for developing and testing new devices, TX or RX strategies, and digital signal processing methods (e.g., coded excitation) [30,33]. Therefore, the proposal of novel dedicated systems based on flexible and cost-efficient solutions is crucial to support the development of new capabilities and features in ultrasound research.

The proposed multifunctional P/R system for forward-viewing IVUS investigation is based on a low-cost FPGA board (USD 423 academic price) and a compact proprietary PCB with state-of-the-art electronic components to control custom transducers according to highly flexible TX and RX schemes. The described digital hardware architecture (TX and RX architecture) is fully controlled by a Matlab-based user-friendly graphical user interface (software architecture), which grants high programmability and flexibility to evaluate simultaneous pulse excitation parameters and RF data acquisition in the same system. The software application can easily control the pulse generator module to match the frequency and bandwidth of the transducer using different window functions for conventional and arbitrary waveform generation. Matlab is chosen because it is one of the most powerful and versatile high-level programming platforms currently available for fast modeling and simulation and the implementation of control and processing algorithms. However, alternative free software platforms can be used, such as LabVIEW Community Edition (National Instruments, Austin, TX, USA) and Visual Studio (Microsoft, Redmond, WA, USA). LabVIEW Community Edition is a free non-commercial-use graphical programming language and development environment created by National Instruments [55]. Microsoft Visual Studio is an Integrated Development Environment (IDE) based on open-source technology and is designed to help developers build a wide range of software applications, including in C++. Examples of customized interface software for the operation of ultrasound research systems designed in C++ and compiled in Visual Studio are described in [33,34].

As shown in Figure 10, the gain voltage controlled by the DAC and two Op Amps (gain of −0.68) needs to be adjusted for accurate measurements. This could be due to the variation in the precision resistors (1%). With a slight adjustment, an excellent agreement between the simulation results and experimental data for the characterization of the RX circuitry is obtained in Figure 11. Additionally, this drift can be compensated for and improved by using components with higher precision.

One of the main challenges of small-dimensional endovascular therapy transducers is generating high acoustic output. During the hydrophone measurements, a pulse duration of 1000 cycles, corresponding to a duration of 1.82 ms at 550 kHz, is applied to a therapy transducer, generating an excitation waveform with a consistent peak amplitude of 72 V. The number of cycles and excitation voltage are determined experimentally under test conditions to avoid damaging either the transducer or the hydrophone. The results reveal that the acoustic output of the 550 kHz transducer is in good agreement with previous work [28]. While higher voltage output would be desirable in the future, the current voltage output is sufficient for intravascular thrombolysis. For example, our group developed a transducer with a 3 mm diameter acoustic metamaterial lens with an enhanced sensitivity of 0.048 MPa/Vpp [28]. In this case, the system can produce a maximum peak negative acoustic pressure of 4.32 MPa with an input voltage of 90 Vpp, i.e., half of the maximum excitation voltage, which is more than sufficient to dissolve a blood clot, especially a red-blood-cell-dominant clot [56,57]. Therefore, this excitation scheme can be helpful in studies of endovascular therapy (sonothrombolysis) for the efficient dissolution of blood clots [28,57]. Another potential application is evaluating the intravascular intensity of the acoustic radiation force impulse (ARFI) imaging to assess the mechanical properties of plaques and arterial vessel walls using displacement tracking, which remains an active area of research [58].

Considering that the strategy for transducer excitation is critical [34,42], the proposed system can generate high-voltage, three-level bipolar waveforms suitable for ultrasound research applications. Testing results show that the proposed system can be easily controlled to match the frequency and bandwidth required for different IVUS transducers across a broad class of applications. As demonstrated in Section 3.2, long multicycle pulses can be applied with high energy transmission for therapeutic ultrasound [28]. As subsequently shown in Section 3.3, high-voltage and high-frequency short pulses can be produced to improve SNR for a high image quality [59]. Additionally, arbitrary pulses can be generated for coded excitation imaging. As discussed by Qiu et al. [34], although unipolar pulsers are widely used in ultrasound systems for their simplicity, unipolar pulses contain more low-frequency out-of-band energy. This limitation restricts its application for many image-enhancing techniques, such as pulse inversion and coded excitation. Alternatively, the tunability of bipolar pulsers leads to the optimization of the transmission energy at the pulse frequency.

Although the presented system, which is designed to be portable and reconfigurable for IVUS imaging and therapy, currently supports only a single-channel pulser/receiver configuration, the FPGA’s inherent parallelism, scalability, and flexibility can be exploited to extend the presented system to multichannel architectures for multi-element transducers. Approximately half of the HSMC connector pins from the FPGA board are available for this extension, which would provide higher image quality and increased acoustic output for high-intensity focused ultrasound. Since the HV7361 pulser requires only three pins for control, parallel pulsers can easily be implemented and independently configured with phase adjustment and arbitrary waveform control to support the development of novel forward-viewing miniaturized IVUS array transducers. A fully integrated multichannel AFE chip can be used for the RX segment due to its low power and area requirements, such as the AFE5805 device (Texas Instruments, Dallas, TX, USA). The AFE5805 is an eight-channel front-end with a programmable low-noise amplifier (LNA), a VCA, a programmable gain amplifier (PGA), a low-pass filter (LPF), and a 12-bit 50 MSPS ADC, which has already been evaluated by our research group [60]. The low-voltage differential signaling (LVDS) data outputs of the ADC reduce the number of interface pins to the FPGA board, enabling high system integration. If additional processing power is needed for sophisticated signal processing techniques and beamforming applications, an Altera DE4 FPGA board (Terasic Inc., Hsinchu, Taiwan) that features a high-end Stratix IV GX EP4SGX230 device (Intel Corporation, Santa Clara, CA, USA) and the same 172-pin HSMC connector can replace the DE2-115 FPGA board.

While IVUS is an established and widely used imaging modality, new approaches to improve image quality and in turn enhance risk assessment continue to be developed. In recent years, many novel transducer technologies for IVUS have emerged; however, experimental testing of these transducers requires ultrasound systems with open and reprogrammable architecture to implement new pulsing techniques, which are not typically available to researchers using commercial systems. For example, the complete dual-channel real-time imaging system designed by Liu et al. [40] to improve NURD artifacts in IVUS is based on a complex FPGA-based board containing two sets of pulser/receivers. In a second study by the same group [41], a novel coded excitation IVUS platform for ultra-high frequency applications was developed. It was demonstrated, with a 121.5 MHz transducer, that the system could generate short and chirp-coded pulses, in which the coded method demonstrated improved SNR at a similar spatial resolution. The system’s main component was a PCB, which had 12 layers of printed circuitry with integrated AFE, FPGA, DAC, OPA, and USB modules. However, since the pulser was specifically designed for high-frequency ultrasound imaging (i.e., >20 MHz), its application with new transducers for low-frequency therapeutic ultrasound is limited.

Considering other systems that have been previously reported [33,34,37], new features can be incorporated in the future. For example, the system can be updated to allow sequences of different excitation waveforms to be programmed, synthesized, and saved into concatenated FPGA LUTs. Then, the LUTs could be scanned sequentially, one for each PRF. In particular, this scheme could be exploited to improve vascular and tissue harmonic imaging using pulse inversion techniques [61,62]. The use of analog switches introduces the possibility of other implementation strategies, including excitation strategies for dual-frequency transducers applied for contrast-enhanced ultrasound imaging [22,63,64,65] or sequences for shear wave elasticity imaging (SWEI) with multiple track beams to track the shear wave generated from a single push (ARFI excitation) [26,58]. The same approaches can be used with the evaluated coded excitation techniques (chirp-coded, Golay sequence, and Barker code) to effectively transmit more energy into the region of interest to enhance the penetration depth by improving the SNR without increasing the peak transmitted power [42,44,45,46,47,48,49].

## 5. Conclusions

A compact, reconfigurable, open multifunctional P/R system specifically designed for new forward-viewing IVUS imaging and therapy has been presented. The proposed system is based on a programmable digital hardware architecture suitable to support the evaluation and characterization of custom IVUS transducers for therapy and imaging applications. The center-aligned PWM technique has been efficiently implemented to synthesize conventional and coded excitation pulses by exploiting the high parallelism of the FPGA to control the generation and transmission of complex three-level arbitrary waveforms with fast RTZ. High-voltage pulsed signals can be produced to excite ultrasound transducers from 500 kHz to 25 MHz with adequate bandwidth and a TX resolution of 200 MHz. The typical TX and RX parameters for ultrasound research, including waveform profile, center frequency, timing, and windowing, have been incorporated into the software interface, allowing for easy and flexible operation during pulse-echo testing. In addition, the system allows direct access to raw RF data with a sampling rate of up to 105 MSPS, increasing its versatility. The use of the system is not limited to the examples described here for validation. With further development, the system could potentially help other research laboratories evaluate innovative excitation and processing approaches, including programmable sequences of pulse inversion [48,49], ARFI excitation [44,45,58], and pulse compression techniques [39,41,44].

## Figures and Tables

**Figure 1 sensors-25-04599-f001:**
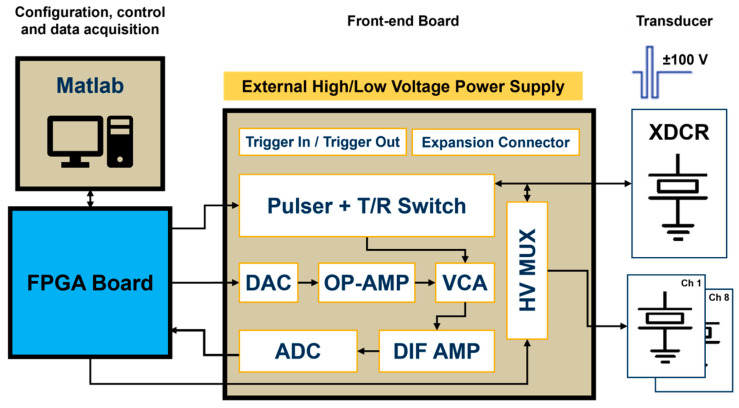
Block diagram of the reconfigurable FPGA-based system for IVUS investigation.

**Figure 2 sensors-25-04599-f002:**
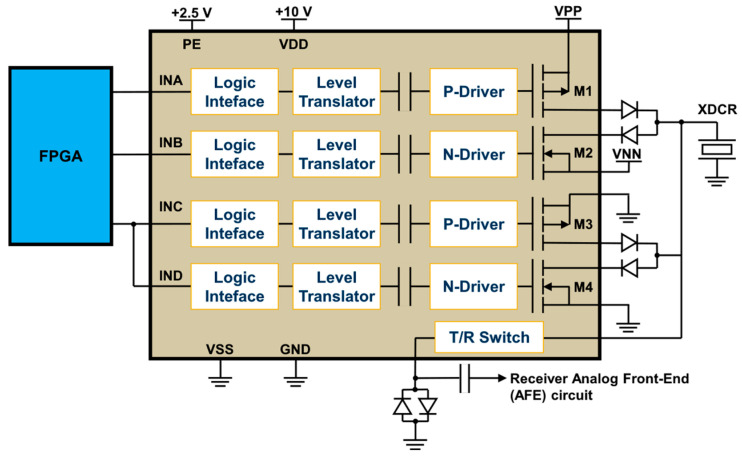
Application circuit of the three-level HV7361 ultrasound transmit pulser. The input pins of the HV7361 pulser are connected to the FPGA, which controls the generation of high-voltage output waveforms to be applied to the transducer (XDCR).

**Figure 3 sensors-25-04599-f003:**
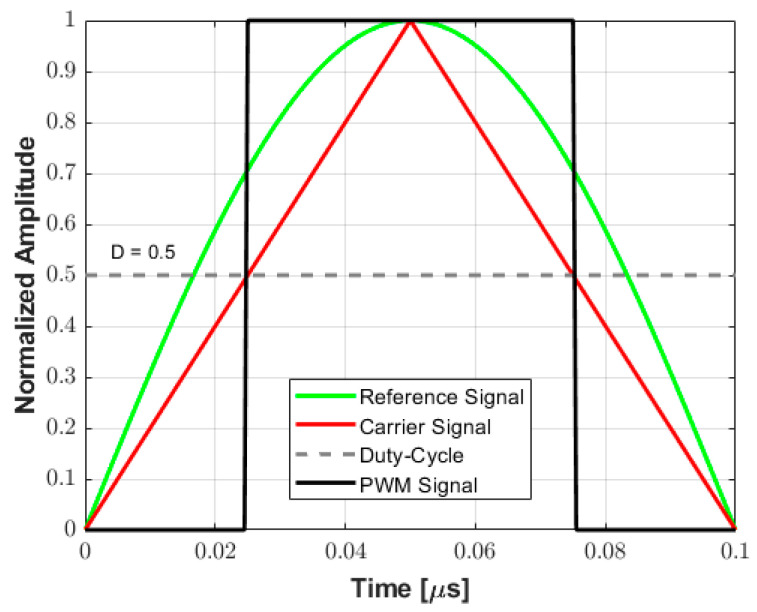
Example of a PWM encoding scheme for a 5 MHz signal. The center-aligned triangle carrier is compared to the rectangular PWM output signal, which is aligned to the peak of the original reference signal. The PWM pulse width is determined by comparing the carrier signal and the duty cycle (*D*) value.

**Figure 4 sensors-25-04599-f004:**
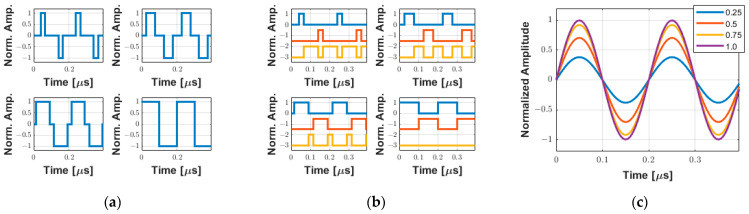
Simulation of two-cycle 5 MHz symmetrical PWM coding using a single pulse per half-cycle with the duty cycle varying from 0.25 to 1.0 in 0.25 increments. (**a**) Tristate-normalized waveforms. (**b**) Input pulse pins INA (blue), INB (red), and INC (orange) used to control the positive, negative, and zero levels, respectively. (**c**) Equivalent behavior of the normalized fundamental frequency component based on the duty cycle variation.

**Figure 5 sensors-25-04599-f005:**
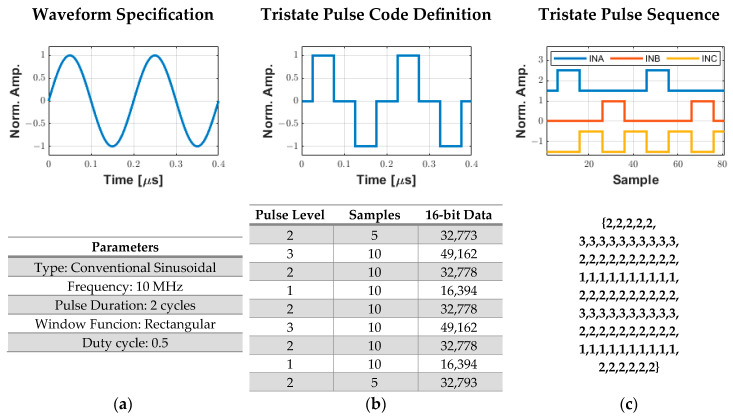
Process for generating digital excitation signals for the ultrasound pulser. (**a**) User specification of the excitation parameters. (**b**) Tristate pulse code data sequence with a 200 MHz resolution that is compressed and stored in a 16k word array to be sent to the FPGA. (**c**) Decompressed output tristate pulse data sequence from the pulse code definition to obtain the INA (level 3), INB (level 1), and INC (level 2) signals.

**Figure 6 sensors-25-04599-f006:**
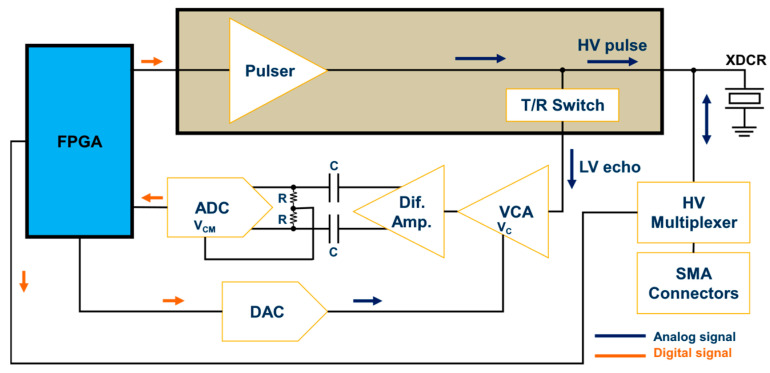
Ultrasound receiver AFE block diagram.

**Figure 7 sensors-25-04599-f007:**
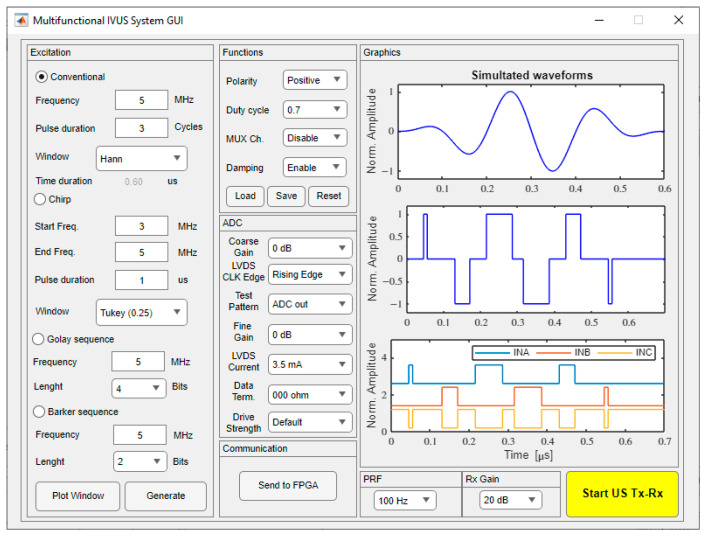
Sample screenshot of the graphical user interface for a three-cycle 5 MHz sinusoidal pulse with a Hanning window and a duty cycle of 0.7.

**Figure 8 sensors-25-04599-f008:**
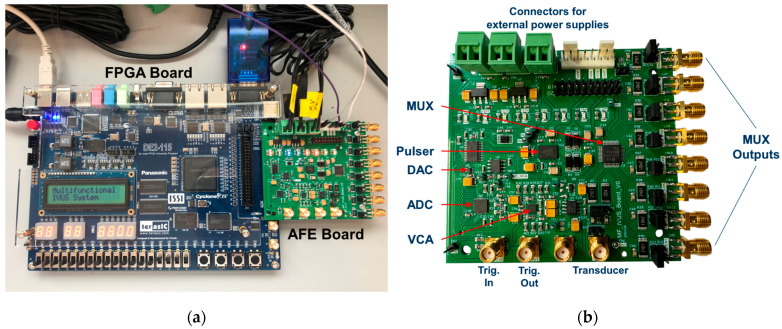
Photos of the hardware system. (**a**) The Altera DE2-115 FPGA control board connected to the AFE board. (**b**) The AFE board with detailed labeling of the main blocks.

**Figure 9 sensors-25-04599-f009:**
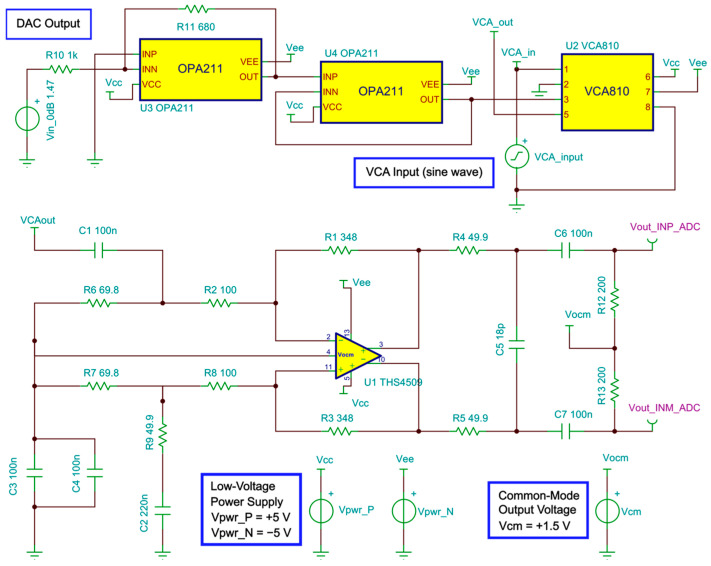
Schematic circuit used to simulate the gain control and frequency response.

**Figure 10 sensors-25-04599-f010:**
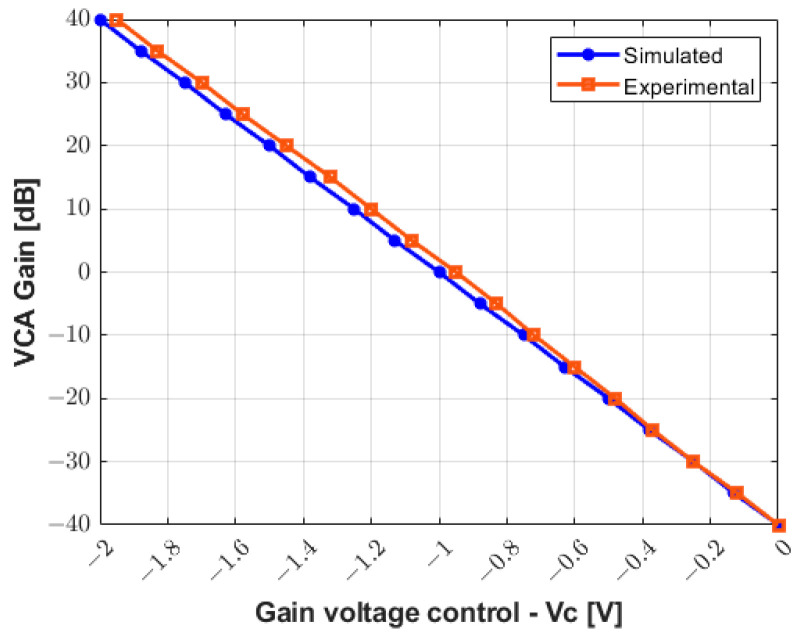
Simulated and compensated experimental results for gain control from −40 dB to +40 dB.

**Figure 11 sensors-25-04599-f011:**
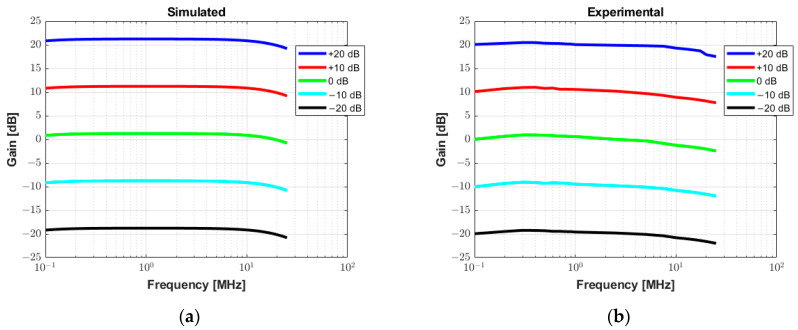
Small signal frequency response analysis with varying gain from −20 dB to +20 dB. (**a**) Simulated AC transfer characteristics in Tina TI software. (**b**) Experimental measurement results.

**Figure 12 sensors-25-04599-f012:**
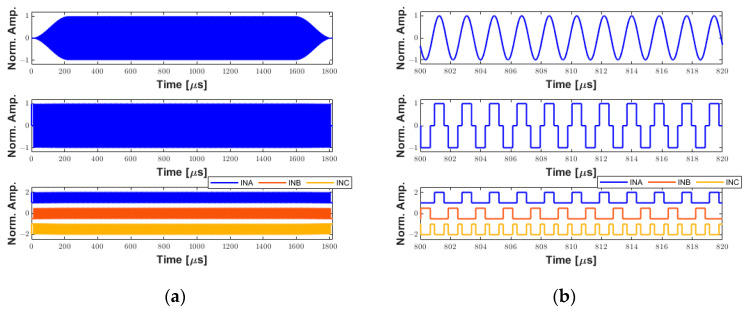
Example of simulated waveforms for therapeutic ultrasound applications. (**a**) A 1000-cycle 550 kHz sinusoidal waveform with a Tukey window (upper panel). Tristate-driven signal with a duty cycle of 70% (middle panel). The respective input control signals to the ultrasound pulser (lower panel). (**b**) A zoom-in on (**a**) from 800 µs to 820 µs for better visualization.

**Figure 13 sensors-25-04599-f013:**
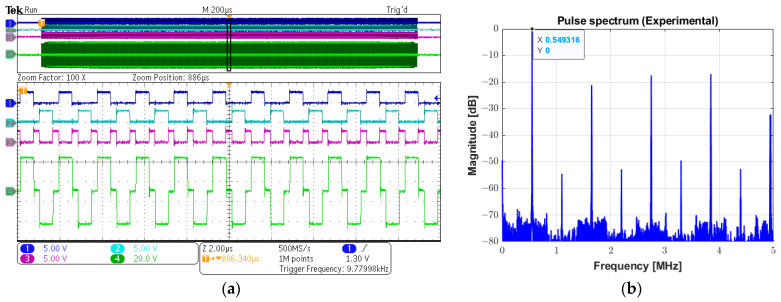
Experimental long pulse for therapeutic ultrasound. (**a**) Input control signals INA (channel 1), INB (channel 2), and INC (channel 3), and the 1000-cycle three-level (channel 4) (upper screen panel), which are zoomed in over a period of 20 µs for a detailed view (lower screen panel). (**b**) Calculated magnitude spectrum of the excitation pulse (channel 4 in (**a**)).

**Figure 14 sensors-25-04599-f014:**
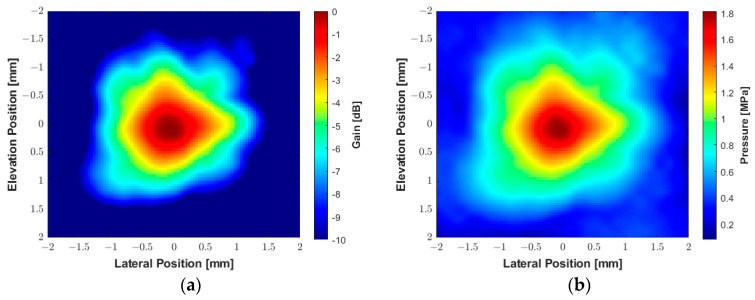
Experimental acoustic maps generated by the custom 550 kHz transducer measured by a hydrophone with an excitation voltage of 70 V and duration of 1.82 ms (1000 cycles). (**a**) The spatial distribution of acoustic gain. (**b**) The spatial distribution of peak negative acoustic pressure. (**c**) Contour beam plot of ultrasound intensity *I_SPTA.3_*. (**d**) Contour beam plot of ultrasound intensity *I_SPPA.3_*.

**Figure 15 sensors-25-04599-f015:**
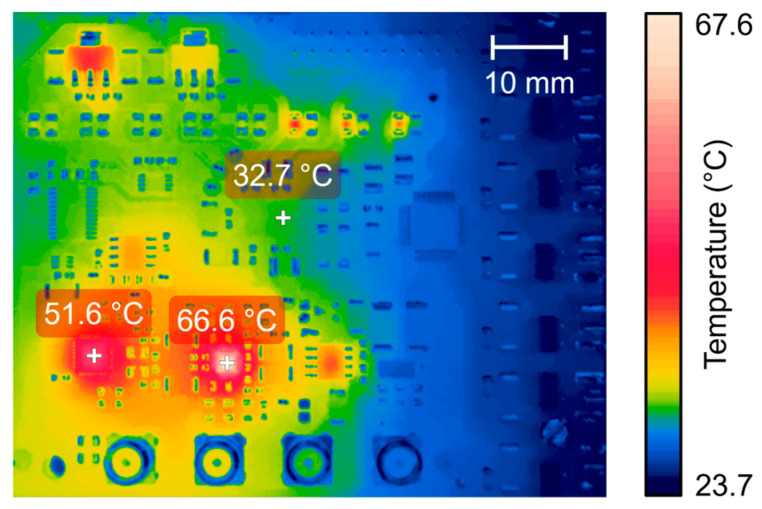
Thermographic image of the proposed AFE board acquired by an FLIR T540 thermal imaging camera with long-duration excitation pulses at the steady-state temperature.

**Figure 16 sensors-25-04599-f016:**
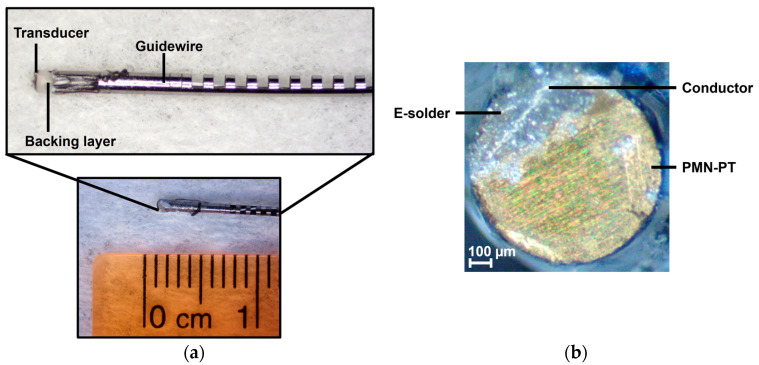
Photos of the guidewire-based high-frequency transducer used for imaging. (**a**) Side view of the 0.88 mm diameter transducer encased in a guidewire. (**b**) Top view of the transducer showing the electrical connection via E-solder.

**Figure 17 sensors-25-04599-f017:**
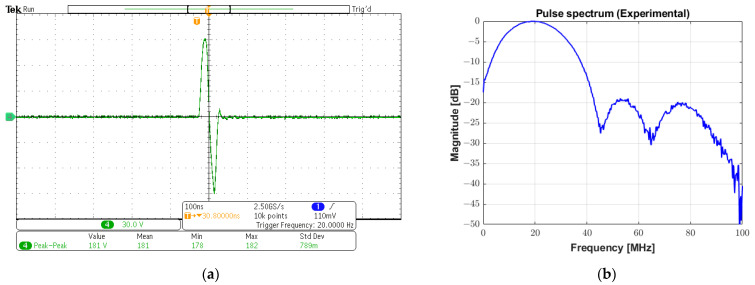
High-voltage bipolar pulse for ultrasound imaging. (**a**) A monocycle 20 MHz bipolar pulse with 181 Vpp generated by conventional excitation with a duty cycle of 100% and rectangular windowing. (**b**) Its spectrum with a −6 dB absolute bandwidth of 28.2 MHz.

**Figure 18 sensors-25-04599-f018:**
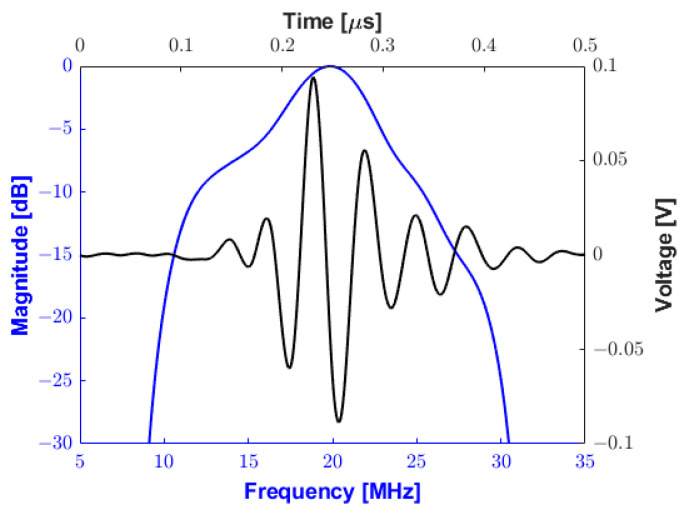
Pulse-echo testing waveform (black line) and frequency response (blue line) using a 75 µm diameter tungsten wire at a depth of 5 mm.

**Figure 19 sensors-25-04599-f019:**
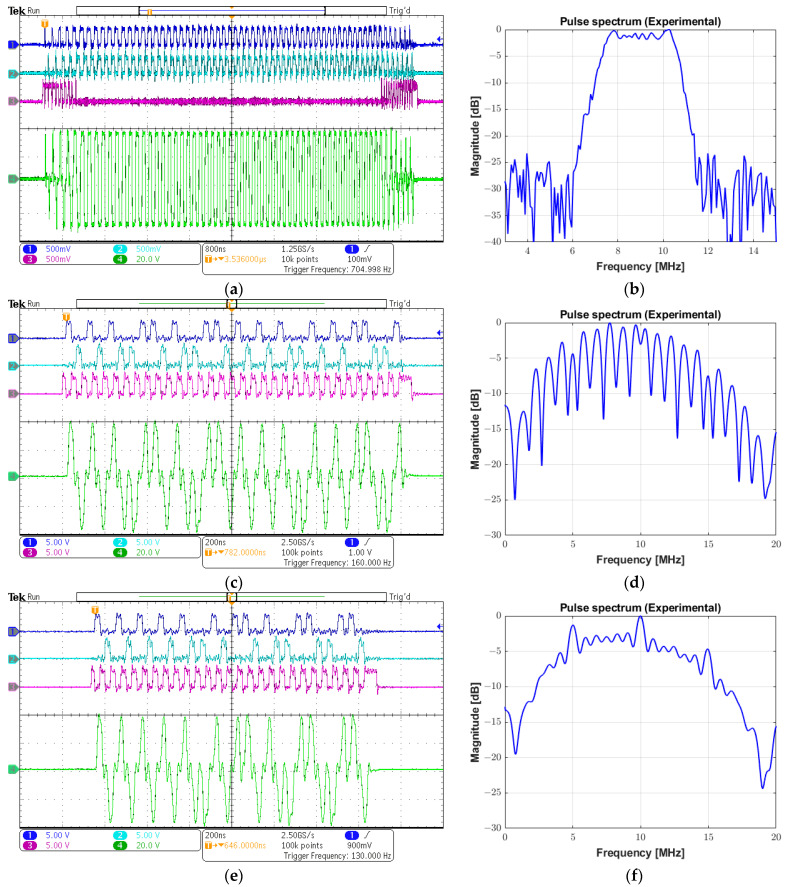
Examples of arbitrary waveforms generated by the system. (**a**) Input control signals and output chirp pulse with a frequency range from 7 to 11 MHz, a Tukey window, and a 7 µs duration. (**b**) Spectrum from (**a**). (**c**) Input control signals and a 16-bit Golay sequence with a central frequency of 10 MHz and a duty cycle of 50%. (**d**) Spectrum from (**c**). (**e**) Input control signals and a 13-bit Barker code with a central frequency of 10 MHz and a duty cycle of 50%. (**f**) Spectrum from (**e**).

**Figure 20 sensors-25-04599-f020:**
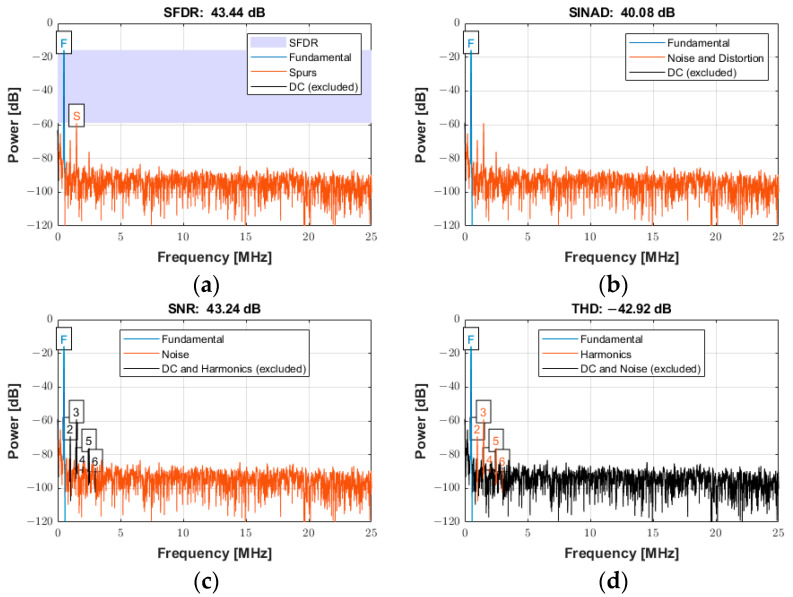
Power spectrum used for quantifying the ADC performance with an input signal frequency of 550 kHz sampled at 100 MSPS for (**a**) SFDR, (**b**) SINAD, (**c**) SNR, and (**d**) THD.

**Figure 21 sensors-25-04599-f021:**
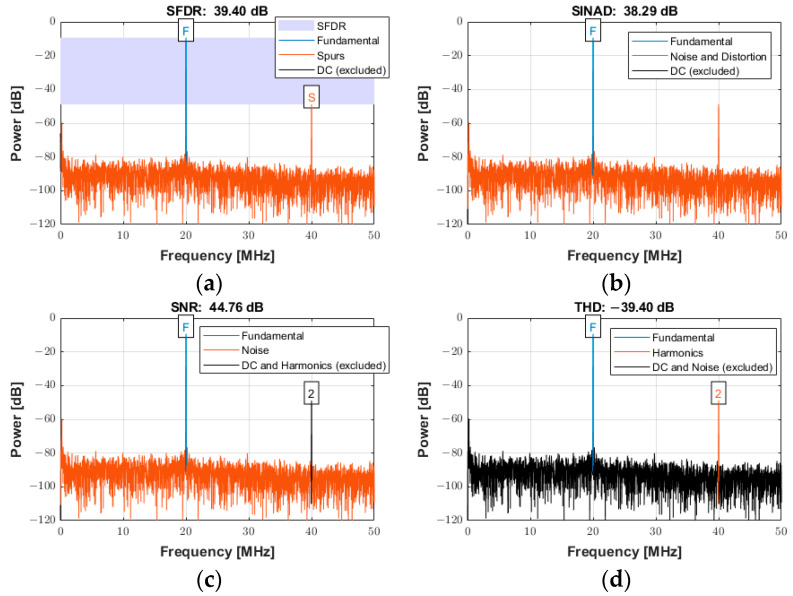
Power spectra used for quantifying the ADC performance with an input signal frequency of 20 MHz sampled at 100 MSPS for (**a**) SFDR, (**b**) SINAD, (**c**) SNR, and (**d**) THD.

**Table 1 sensors-25-04599-t001:** HV7361 pulser logic control table.

PE ^1^	Input Pulse Pins				Output MOSFETs				Excitation Voltage
	INA	INB	INC	IND	M1	M2	M3	M4	
0	X	X	X	X	OFF	OFF	OFF	OFF	-
1	1	0	0	0	ON	OFF	OFF	OFF	VPP
1	0	1	0	0	OFF	ON	OFF	OFF	VNN
1	0	0	1	0	OFF	OFF	ON	OFF	0
1	0	0	0	1	OFF	OFF	OFF	ON	0

^1^ The PE pin enables the logic interface and is used as a voltage reference input (+2.5 V) to the FPGA.

**Table 2 sensors-25-04599-t002:** Specification of the 16-bit array used to save the duration for each state of the pulse.

Level Definition	Signal	Field (2 Bytes)		
		Bit 15	Bit 14	Bit 13-0
0	Disable	0	0	Irrelevant
1	INB	0	1	Number of negative samples
2	INC/IND	1	0	Number of zero (damping) samples
3	INA	1	1	Number of positive samples

**Table 3 sensors-25-04599-t003:** Specifications of the high-frequency imaging transducer.

Property	Description
Piezoelectric Material	PMN-PT
Center Frequency	19.48 MHz
Bandwidth (−6 dB)	38.59%
Element Diameter	0.88 mm
Backing Material	Loaded epoxy (Epotek 301 and aluminum oxide)
Matching Layer(s)	None

**Table 4 sensors-25-04599-t004:** FPGA utilization summary (Cyclone IV E EP4CE115F17C6N FPGA).

FPGA Resource	Utilization	Available	% Utilization
Logic Element (LE)	7090	114,480	6
Register (FF)	4384	117,053	4
Look-Up Table (LUT)	539	7155	8
Embedded Memory (Bits)	583,808	3,981,312	15
DSP Blocks (Multiplier)	4	532	<1
Phase-Locked Loop (PLL)	2	4	50

## Data Availability

Dataset available on request from the authors.

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
