# Peer review of "An Open Multifunctional FPGA-Based Pulser/Receiver System for Intravascular Ultrasound (IVUS) Imaging and Therapy"

_sensors, 2025, doi:10.3390/s25154599_

Round 1

Reviewer 1 Report

Comments and Suggestions for Authors
  1. The Introduction and Discussion sections would benefit from a broader review of recent IVUS platforms. For example, studies on 100 MHz high-frequency ultrasound systems supporting single-pulse and coded excitations, and dual-channel IVUS designs addressing non-uniform rotational distortion (NURD), are relevant. Please include such references, discussing their advantages and challenges to better highlight this system’s innovations.
  2. The 20 MHz transducer testing (Section 3, Figure 16) lacks physical description. Authors should include a photograph of the transducer and a table listing key parameters.
  3. Figures 1 and omit the two-stage amplifiers (OPA211, THS4509) following the DAC in the receiver chain, Authors should update these figures to accurately depict the receiver architecture.
  4. The receiver chain description lacks details on filtering to mitigate high-frequency noise and prevent ADC aliasing. Authors should clarify if filters are implemented (e.g., anti-aliasing filters) and how they address noise, or justify their absence.
  5. The font size in Figure 16 is disproportionately large compared to other figures. Please reduce it to ensure consistency with the manuscript’s visual style.

Reviewer 2 Report

Comments and Suggestions for Authors

The manuscript focuses on the technical development of the FPGA-based pulser/receiver system. My main concern is about the novelty of this study. The authors should address it clearly in the manuscript. 

Other comments: 

  1. The system’s high-voltage power supply is limited to ±90 V, which may restrict its ability to achieve the full potential of high-voltage excitation for certain therapeutic applications. The authors should address this.
  2. The paper does not address the thermal performance of the system during prolonged use, particularly when generating long-duration pulses. Overheating could degrade component performance or even damage the system. The authors should address this. 
  3. The authors suggest to discussing the system’s noise floor and signal integrity under different operating conditions. 
  4. The system currently supports only a single-channel pulser/receiver configuration. The paper does not explore scalability to multi-channel setups, which are often required for advanced imaging techniques. 
  5. The system relies on MATLAB for the GUI and control interface, which may limit accessibility for users who do not have access to MATLAB licenses. 

Round 2

Reviewer 2 Report

Comments and Suggestions for Authors

The authors have properly addressed all of my comments. I have no further comments.